# Construction and Synthesis of MoS₂/Biocarbon Composites for Efficient Visible Light-Driven Catalytic Degradation of Humic Acid

Chencheng Wang [1,2,†], Ning Wang [3,†], Huicheng Ni [4], Congcong Yao [1], Junchao Qian [1,*], Jianqiang Wei [1], Jianping Chen [5] and Zhiren Wu [2,*]

1 Jiangsu Key Laboratory for Environment Functional Materials, Suzhou University of Science and Technology, Suzhou 215009, China
2 School of the Environment and Safety Engineering, Jiangsu University, Zhenjiang 212013, China
3 School of Materials Science and Physics, China University of Mining and Technology, Xuzhou 221116, China
4 School of Environmental and Chemical Engineering, Jiangsu University of Science and Technology, Zhenjiang 212003, China
5 Jiangsu Key Laboratory of Intelligent Building Energy Efficiency, Suzhou University of Science and Technology, Suzhou 215009, China
* Correspondence: ziyou1900@gmail.com (J.Q.); wuzhiren@ujs.edu.cn (Z.W.); Tel.: +86 18013197989 (J.Q.)
† These authors contributed equally to this work.

**Abstract:** MoS₂/bio-template carbon composite materials with outstanding photocatalytic degradation performance were constructed and synthesized by an impregnation–hydrothermal–calcination (IHC) method. Composites of the same type were synthesized by a direct-impregnation–calcination (DIC) method for comparison. The results showed that calcination process was obtained from biotemplate carbon with preserved structure. IHC method obtained petal-like MoS₂, while DIC method obtained needle-like MoS₂. The composite material exhibits adsorption–catalytic degradation performance. Driven by visible light, the photocatalytic degradation efficiency of the composites synthesized by IHC method for humic acid reached 98.73% after 150 min of illumination.

**Keywords:** MoS₂; bio-template carbon; composite; degradation; humic acid





## 1. Introduction

As a typical type of two-dimensional layered transition metal dichalcogenides (TMDs) semiconductors, molybdenum disulfide (MoS₂) and corresponding composite materials are applied to various fields, such as effective visible light photocatalysis [1,2], highly sensitive sensor [3,4], novel energy storage [5,6], and so on [7]. The microscopic atomic arrangement of two-dimensional MoS₂ nanophase materials presents a sandwich structure, which means that in a single-layer MoS₂ with three atomic layers, the Mo atoms occupy the middle layer, while the upper and lower two layers are, respectively, filled with S atoms. The Mo and the S atomic layers are connected by covalent bonds [8]. For multilayer MoS₂, MoS₂ monolayers connect with each other by Van der Waals forces, and the layer spacing is approximately 0.65 nm [9]. Due to the distinct micro-geometries [10,11] and splendiferous properties [12], two-dimensional MoS₂ materials exhibit large specific surface areas [13], good electrical conductivity, strong adsorption properties [14], and possess high activity sites [15]. Thus, this type of material has been extensively investigated by researchers in recent years. However, when used as a photocatalyst, a single MoS₂ is highly prone to exhibit low photocatalytic activity due to the rapid recombination of photogenerated carriers caused by the limitation of the energy band structure. Therefore, it is generally necessary to compound MoS₂ with other types of materials to enhance the photo-responsive performance of the catalyst.

The construction of composite systems of sulfides and carbon materials is an effective way to enhance the photocatalytic performance of $MoS_2$. The most commonly used carbon materials include graphene [16], graphene oxide, reduced graphene oxide [17], carbon nanotubes [18], and carbon fibers [19,20]. Lee et al. synthesized $MoS_2$/graphene hybrid materials through the microwave hydrothermal method. The composite materials changed the charge transfer resistance and photocurrent intensity [21]. The maximized photocatalytic hydrogen production efficiency under optimized conditions was able to reach 667.2 $\mu mol \cdot h^{-1} \cdot g^{-1}$. In addition to photocatalytic hydrogen production studies, sulfide/graphene oxide composites have also shown promising potential in the field of photodegradation [22]. The modified $MoS_2$@graphene oxide (GO) material is able to realize the total photo-oxidative desulfurization of thiophene within 75 min driven by visible light. The composite system has a good recyclability, which increases the lifetime of the photocatalyst [23,24]. Although the composite process with carbon materials can effectively enhance the catalytic performance of sulfides, graphene and related materials have problems, such as high synthesis cost, and safety hazards in mass preparation process, especially the high surface energy of graphene, makes it prone to curling and thus hinders the subsequent loading of $MoS_2$. To address the above-mentioned problems of carbon materials, we propose the utilization of a bio-template carbon material obtained by high-temperature adiabatic heating using plants as templates, so as to achieve similar photocatalytic performance enhancement effects.

Humic acid is a macromolecular organic substance that has been widely present in the natural world since and is capable of promoting increased food production [25]. Humic acid can effectively improve the soil structure, accelerate the transfer of nutrients to crops, and chelate trace metal elements suitable for plant growth. However, when humic acid gets into water bodies due to rainwater scouring or soil erosion, it not only decreases the content of metal ions and trace elements in water but also affects the toxicity and biological effectiveness of metal ions [26]. More importantly, humic acid-like substances in water bodies are important precursors of halogenation by-products. Humic substances are highly susceptible to the formation of disinfection by-products (DBPs) and trihalomethane-like carcinogens (THMs) during the chlorination process in water plants [27]. For these reasons, the researchers propose to use sulfide semiconductor materials that can degrade organic matter driven by visible light to deal with the environmental problem of humic acids in water bodies, suggesting a novel application idea for photocatalytic semiconductors in the field of environmental purification.

Considering the structural properties of leaves and crystalline characteristics of $MoS_2$, the impregnation–hydrothermal–calcination (IHC) method was selected to synthesize $MoS_2$/bio-template carbon composites. To investigate the effect of the choice of synthesis process on the morphology and properties of the material, a direct-impregnation–calcination (DIC) method was introduced into this research.

In this work, The $MoS_2$/bio-template composites with visible light response were constructed to achieve the rapid degradation of humic acid in a short time. Biocarbon was introduced to promote the separation of photogenerated carriers in $MoS_2$ and increase the adsorption capacity of the composites in the catalytic degradation experiments. The physical phase analysis further verified the differences caused by the preparation methods, mainly in the positions and intensities of the crystal characteristic peaks. The experimental results provide a potential possibility for photocatalytic semiconductors in practical production.

## 2. Results and Discussion

### 2.1. Determination of Calcination Temperature

Before the mass production of materials, the thermogravimetric analysis–differential scanning calorimetry test was used to determine the calcination temperature of these two synthesis methods. The two dried precursors were, respectively, heated to 800 °C at the rate of 10 °C·min$^{-1}$ in an atmosphere of $N_2$ to observe the mass changes of the precursor,

as well as energy changes of the system from room temperature to 800 °C. The precursors tested for the DIC method were dried leaves after immersion (Figure 1a), whereas the samples tested for the IHC method were leaves after dried hydrothermal (Figure 1b).

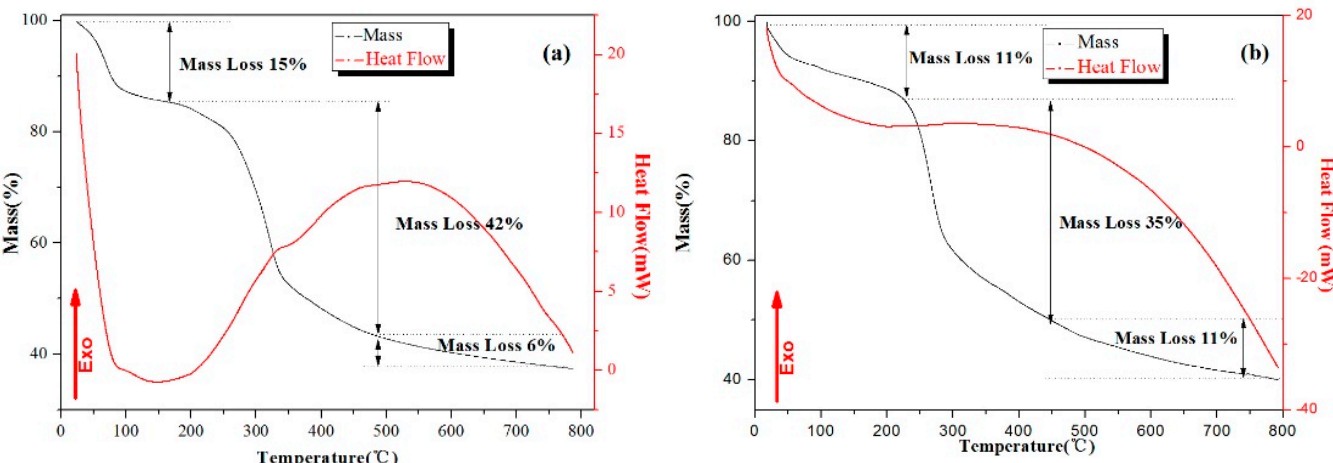

**Figure 1.** TG and DTA curves of the precursor for (**a**) the DIC method and (**b**) the IHC method.

For the process of DIC method, the whole weight loss process can be divided into three main parts. The first stage of weight loss was about 15%, which occurred from the room temperature to 150 °C, mainly including the evaporation of free-water in the leaves (room temperature to 100 °C) and the detachment of bound-water (120 °C to 150 °C). Either the evaporation of free-water or the release of bound-water required endothermics; therefore, the downward trend showed in the heat flow curve. A 42% weight loss consisted of two simultaneous processes, namely the carbonization of organic matter and the formation of part of $MoS_2$. At about 350 °C, a slope transition occurred in both the weight and heat flow curves, corresponding to the carbonization of cellulose (250 °C to 300 °C) and lignin (200 °C to 500 °C), respectively. The reason for the 6% weight loss of the system after 500 °C was the carbonization of the remaining lignin [28].

For the thermal analysis of IHC method, according to the heat flow curve, the weight loss process of the system could be divided into three parts as well. In the dehydration process before 200 °C, the value of weight loss in IHC method was lower than that of the DIC method. The main reason for this change was that the samples in the IHC method are hot-dried at 60 °C after hydrothermal treatment, while the sample in the DIC method was dried at room temperature. After hydrothermal heating, the microscopic surfaces of the fiber, cell wall, and other substances in the leaves were covered by the partially formed $MoS_2$, so that the carbonization of cellulose and lignin required a higher temperature and less heat would be released from the whole system.

To sum up the two figures and make the organics in the precursor efficiently transformed into bio-template carbon, on which $MoS_2$ could be more fully grown, leading to the promotion of photocatalytic performance of $MoS_2$. The calcination temperature was determined as 700 °C.

### 2.2. Micromorphology Analysis of Bio-Template Carbon

For the macroscopic morphology of the flaky leaves of plants, the microstructure was relatively diverse (Figure 2). Both sides of the surface are in the shape of flaky structure, the tubular structure was closely arranged inside the blade and the pipe wall was evenly arranged with uniform geometric concave–convex structure. The reason for the formation of this hybrid structure was mainly because the structure conformed to the growth needs of plants. The large surface area of the surface could maximize the contact with the external environment, which promoted the adsorption of water and the absorption of sunlight; tubular and bumpy structures allowed for the targeted transport of plant-advancing mate-

rial inside leaves. After being carbonized by special means, the morphology of leaves was retained. The microstructure mentioned above was able to provide numerous nucleation sites and active sites during crystal growth, benefiting the rapid formation and growth of the grains. In addition, if the MoS$_2$ nanomaterials could be combined with this type of bio-template carbon to form composites, the original properties of MoS$_2$ might be further improved.

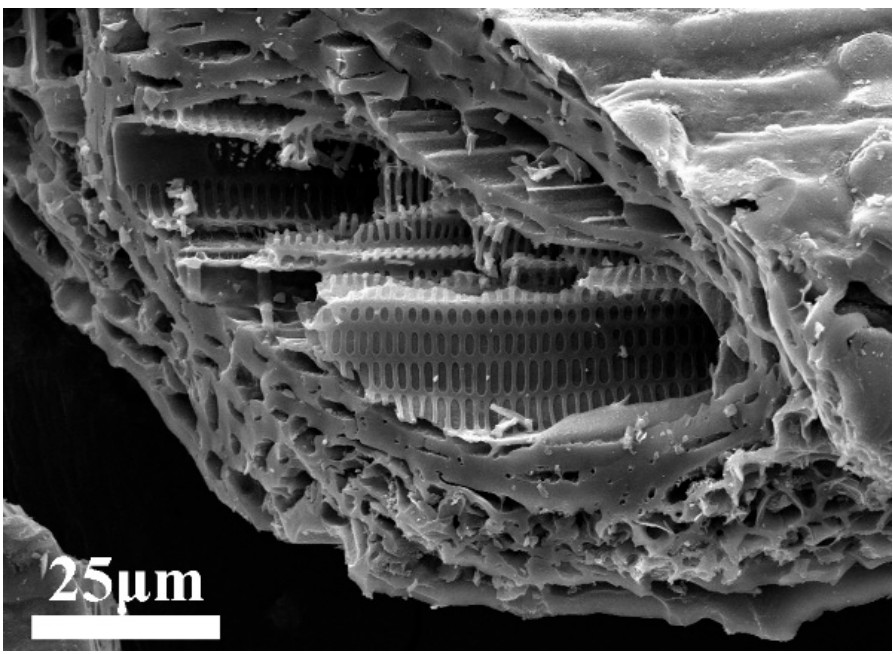

**Figure 2.** SEM image of microstructure morphology for flaky leaves.

*2.3. Micromorphology and Structure Analysis of Composites*

The SEM images of materials obtained by the two methods were displayed in Figure 3. Obviously, the carbon matrix microstructure of the materials prepared by the DIC and IHC methods preserved the micromorphology of the bamboo leaves well. However, the morphology of MoS$_2$ materials that formed on the surface of carbon materials were different. The MoS$_2$ synthesized by the DIC method showed a needle-like structure (Figure 3a,b) which has been rarely reported in research, while MoS$_2$ obtained by the IHC method was in shape of a lamellar stacked flower ball structure (Figure 3c,d). In addition, compared with Figure 3a,c, not only was MoS$_2$ observed on the former but also on the spiny structure on the surface of bamboo leaves, with the latter completely covered by MoS$_2$. Despite both the DIC method and the IHC method possessing impregnation steps and the amount of Mo and S ions adsorbed in the precursor in this process being similar, the IHC method had the hydrothermal process, in which additional Mo ions and S ions were introduced into the solution, resulting in the MoS$_2$ produced by the IHC method being more than the DIC method. In the DIC method, MoS$_2$ was obtained in the process of calcination, and the content was not high enough, so the structure of the sulfide material tends to be in shape of needle, while the IHC method in MoS$_2$ was mainly prepared during the hydrothermal process, and sulfide materials tend to form a lamellar stacking structure. Additionally, the calcination process chiefly converted organic carbon in leaves into bio-template carbon, which hardly change the morphology of already-formed MoS$_2$.

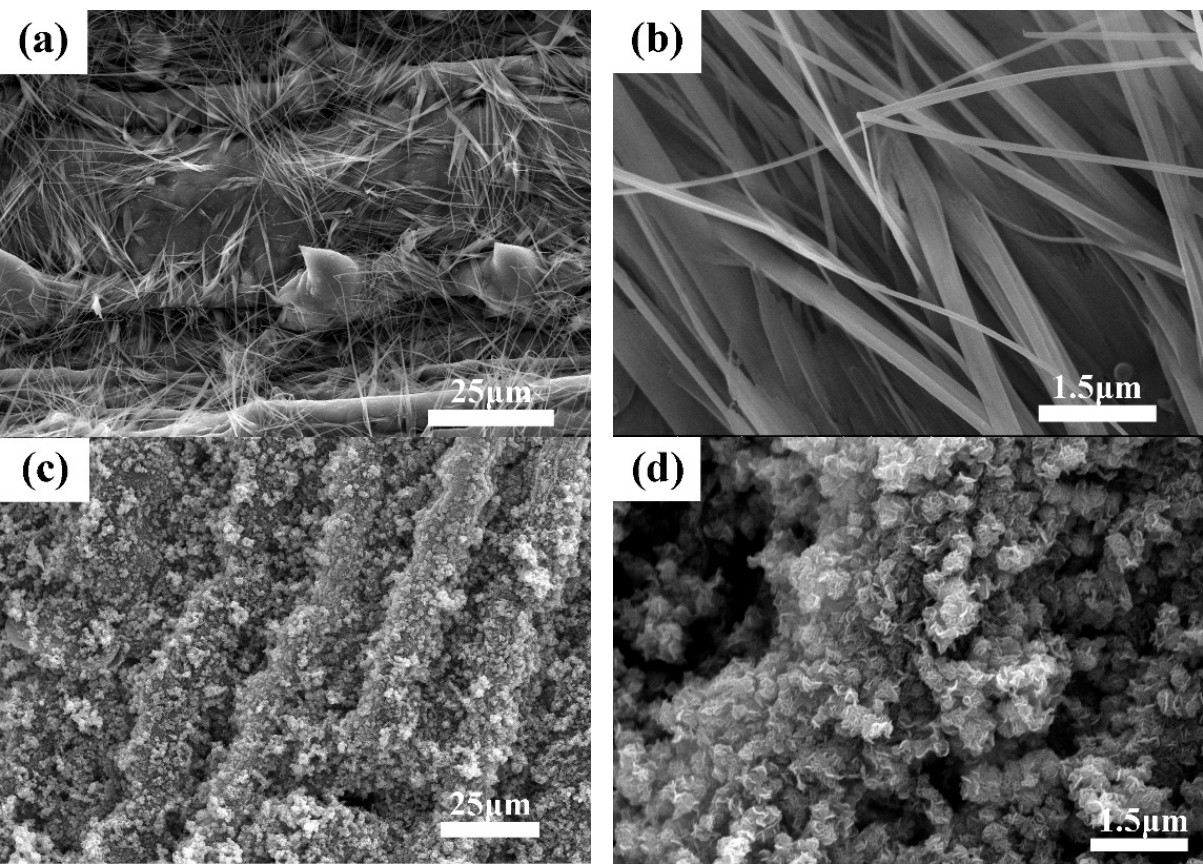

**Figure 3.** SEM images of MoS$_2$/biocarbon obtained by DIC method (**a**,**b**) and by IHC method (**c**,**d**) under different magnification.

Figure 3 displayed SEM figures of MoS$_2$/biocarbon photocatalytic composite materials obtained by the direct-impregnation–calcination (DIC) method and impregnation–hydrothermal–calcination (IHC) method, respectively. The TEM images of the MoS$_2$ obtained by the DIC method are displayed in the Figure 4a, b, and those of the MoS$_2$ obtained by the IHC method are presented in Figure 4c,d. For the DIC method, the needle-like MoS$_2$ material was firmly combined with the carbon matrix, and this impactful combination was able to promote the efficient transfer of free electrons between two kinds of materials. The high-resolution image in Figure 4b clearly reveals the microstructure of the needle-like MoS$_2$ and the interplanar spacing of 0.61 nm, corresponding to the (002) crystal plane of MoS$_2$. Under this magnification, the needle-structure was found to be stacked in order of layered MoS$_2$. The ordered stacking was attributed to the slow heating and long-time heat insulation in the calcining process. However, due to the limited content of two ions, the ordered stacking formed a needle-like structure with a sharp upper and rough bottom to constitute a stable shape, which was consistent with the results of SEM analysis. Compared with the MoS$_2$ in the ordered arrangement prepared by the DIC method, the arrangement of petal-like MoS$_2$ prepared by the IHC method is more disordered. Under the low-resolution condition (Figure 4c), the thin layer of MoS$_2$ extended in all directions and was stacked into the "MoS$_2$ flowers", which was observed in SEM. The reason for the formation was mainly because during the hydrothermal process, most of the Mo and S ions in the solution were bonded to form MoS$_2$ under the action of temperature and pressure and grew on the carbon substrates with high activity sites. Due to the high Mo and S ion content and longer hydrothermal time, MoS$_2$ was able to be fully grown and stacked to form a "bouquet" in the presence of solution. Under the high-resolution condition (Figure 4d), "MoS$_2$ flowers" were more clearly observed. The darker slice was MoS$_2$, which was stacked vertically or nearly vertically to the direction of observation, while the brighter part was the stacked

MoS$_2$, which was parallel or slightly at an angle to the observation direction. According to the results of SEM and TEM analysis, it could be concluded that for composites synthesized from the DIC method and IHC method, respectively, the bonding mode of MoS$_2$ and bio-template matrix, as well as the microstructure of materials, could improve the performance of the composites systems.

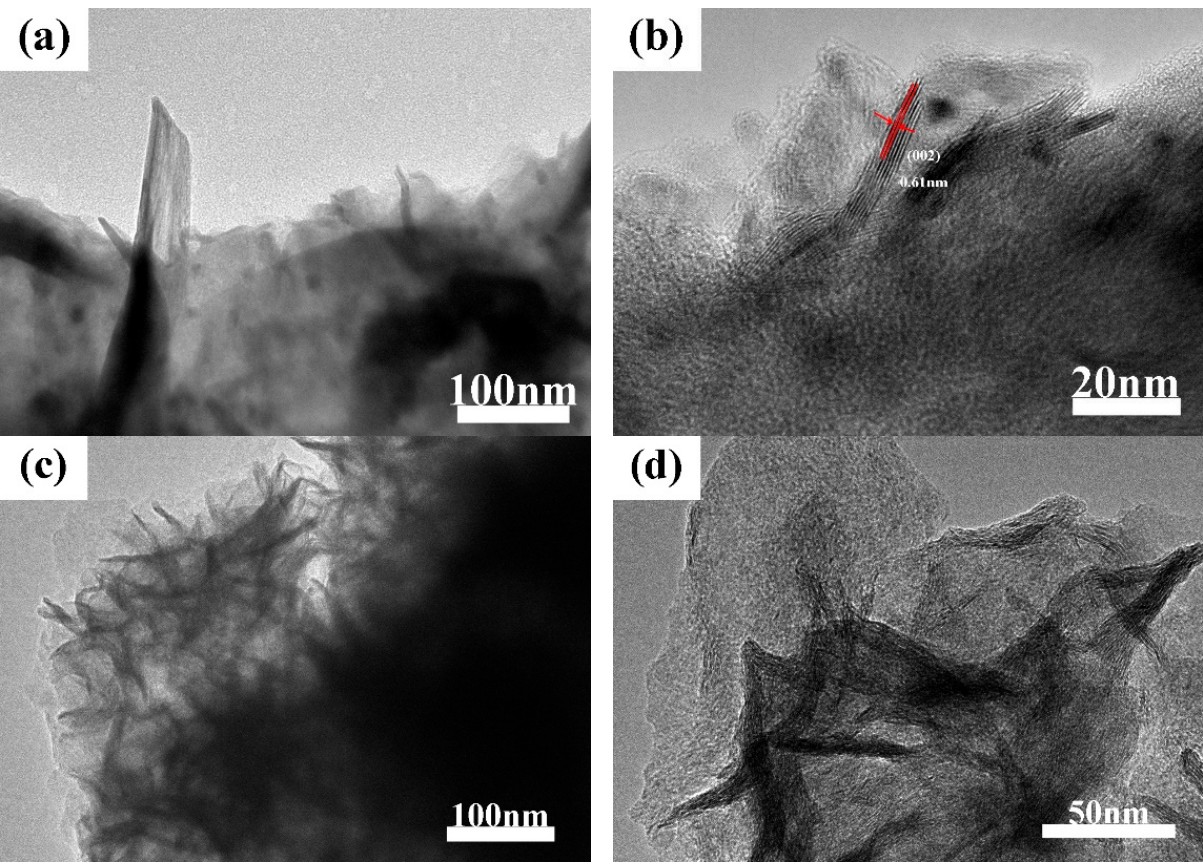

**Figure 4.** TEM images of MoS$_2$/biocarbon obtained by DIC method (**a,b**) and by IHC method (**c,d**) under different magnification.

### 2.4. X-ray Powder Diffraction Phase Analysis

The phase and structural circumstances of composites synthesized by DIC method and IHC method were thoroughly investigated using XRD powder diffraction analysis. The characteristic peaks of two kinds of materials were scanned from 10–80°, as shown in Figure 5. The figure of materials revealed characteristic peaks at 2θ of about 14.4°, 33.5°, 39.4°, 59.2°, 69.8°, and 75.3°, which correspond to diffraction from (002), (101), (103), (008), (108), and (116) crystallographic planes, respectively. These information peaks were found to be matched with JCPDS No. 00-037-1492. It was noteworthy that these peaks were not all in the same XRD material pattern. In addition to the characteristic peaks of the main location, a certain peak appeared at a specific location in one material and may not be observed in the other. The difference demonstrated that different preparation methods result in differences of the final obtained materials in details. Compared with the IHC method, the material obtained by the DIC method had a large number of impurity peaks in the XRD and the intensity of the MoS$_2$ characteristic peak was not high. The additional peaks appearing at 20–30° in the XRD of the materials prepared by the DIC method are mainly caused by the amorphous biocarbon. The peaks appearing at 55–60° are the (110) and offset (008) crystallographic planes of MoS$_2$ [29,30]. The reason for this phenomenon was not only that did MoS$_2$ form during the hydrothermal process but also that the impurity ions in the precursor were able to be dissolved in the solution, so as

to reduce the influence on the material purity. While the above-mentioned process was not available in the DIC method, so other peaks were observed in the XRD pattern of materials obtained by the DIC method. In addition, a bulge appeared at about 25°, which indicated that the material contained amorphous substance. It can be speculated that the amorphous substance was bio-template carbon, which was synthesized after high temperature calcination by referencing relevant research reports and analyzing the related test results. The XRD phase analysis of the material showed that the $MoS_2$/biocarbon composite materials were prepared by both DIC method and IHC method. However, due to the differences in steps of preparation methods, the phase analysis presented details differently.

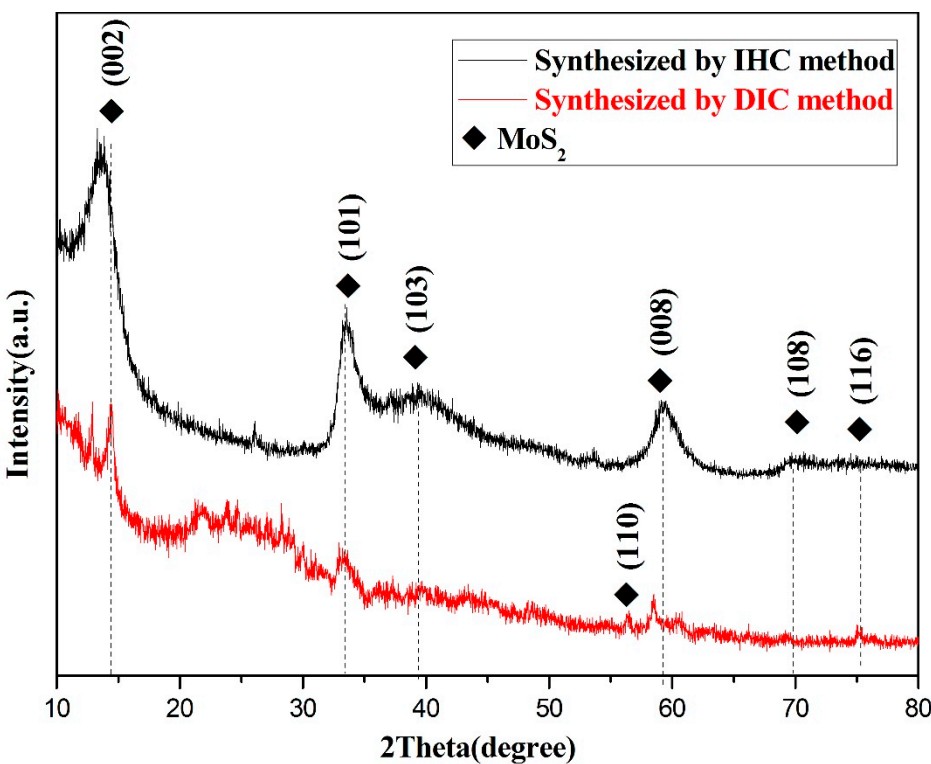

**Figure 5.** XRD patterns of $MoS_2$/biocarbon obtained by DIC method and by IHC method.

### 2.5. Raman Spectra Analysis

Figure 6a displayed the Raman spectra of two materials. The Raman peak of $MoS_2$ appeared before about 500 cm$^{-1}$, while D peak ($A_{1g}$) and G peak ($E_{2g}$) of carbon were located at about 1360 cm$^{-1}$ and 1580 cm$^{-1}$, respectively [31]. The main reason for the appearance of the D peak was that the atomic structure of the carbon material was defective, and the stretching motion of all the sp2 atom pairs in the long chain of the molecule induced the formation of the G peak. At about 2700 cm$^{-1}$, the intensity of the 2D peak was not very clear, indicating that the carbon matrix in the material was amorphous carbon, consistent with the amorphous bulge in XRD analysis. After magnifying the curve between 370 cm$^{-1}$ and 440 cm$^{-1}$ in Figure 6a, the in-plane and out-plane $E_{2g}$ and $A_{1g}$ vibrational modes are displayed in Figure 6b. Although the two kinds of $MoS_2$ belong to the multi-layer materials, the analysis of the TEM revealed that there was a difference in the order of arrangement of the layered $MoS_2$.

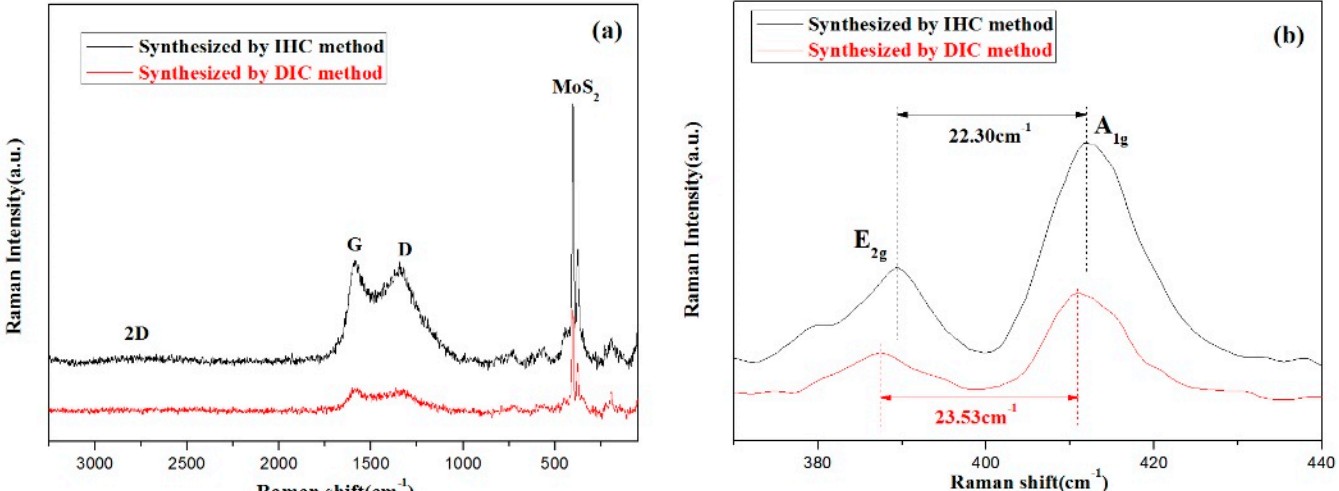

**Figure 6.** Raman spectra of $MoS_2$/biocarbon obtained by DIC method and by IHC method (**a**) enlargement indicating their multilayer characteristics (**b**).

### 2.6. X-ray Photoelectron Spectroscopy Analysis

The products synthesized by the IHC method were subjected to XPS spectroscopy to investigate the elemental valence and bonding patterns in the products. The XPS energy peaks corresponding to Mo $3d_{3/2}$ and Mo $3d_{5/2}$ were located at 232.48 eV and 229.30 eV, respectively, (Figure 7a) after equation fitting. The binding energy difference between the two peaks was about 3.18 eV, indicating that the valence state of the element Mo in the composite was +4. Figure 7b showed the energy spectrum of S 2p in the product. The S 2p energy level was able to be split into two spin orbitals, $2p_{1/2}$ (163.29 eV) and S $2p_{3/2}$ (162.10 eV), which were assigned to $S^{2-}$ in the $MoS_2$ semiconductor. The two fitted peak positions of S 2p matched well with the corresponding binding energy of $S^{2-}$, indicating that $Mo^{4+}$ in the $MoS_2$ semiconductor formed a strong bond with $S^{2-}$ and the sulfide possessed excellent stability, thus the chemical stability of the composite was able to be effectively ensured. For the C element in the product (Figure 7c), the fitted peak representing carbon with hydroxyl groups (C-OH) was located at 286.29 eV, while the fitted peak representing C-C or C=C was located at 284.76 eV. The binding of carbon to hydroxyl groups was mainly the bonding of bio-template carbon to bound water, and a small number of OH groups in solution adsorbed to the surface of the bio-template for stable binding during hydrothermal processes; the single and double bonding between carbon atoms occurred mainly during the conversion of leaves to bio-template carbon, where the carbon chain backbone of organic substances was retained, and the functional groups in the sugars were reorganized under temperature and pressure to form C=C. Figure 7d was obtained by comparing the C 1s in the products synthesized by the IHC method (top) and the simple DIC method (bottom), respectively. In addition to the C-OH binding peak (284.62 eV) and the C-C or C=C binding peak (284.69 eV), a convoluted peak at 289.13 eV was present for element C of the product obtained by the DIC method. The convolution peak was a fitted peak for the carboxyl group (HO-C=O), indicating that the hydrothermal process facilitated the removal of the carboxyl group from the bio-template carbon. Furthermore, when comparing the areas of the C-C or C=C fitted peaks in the two materials, the areas of the C-C or C=C peaks in the products of the IHC method were much larger than those of the DIC method, indicating that the combination of the calcination and hydrothermal processes was more conducive to the efficient conversion of original bio-template to bio-template carbon.

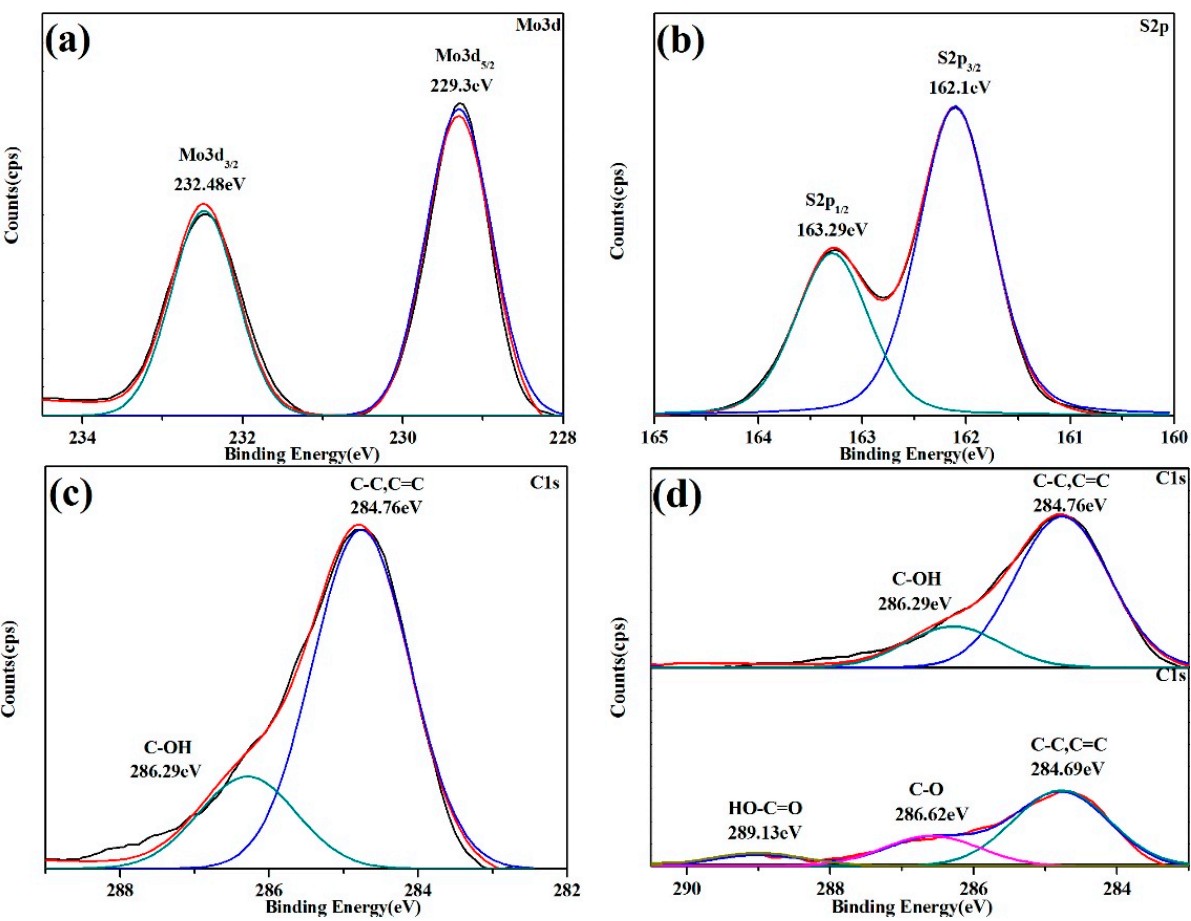

**Figure 7.** XPS spectra of (**a**) Mo 3d core-level structure, (**b**) S 2p core-level structure, (**c**) C 1s core-level structure, and (**d**) comparison of C1s in the products synthesized by the IHC method (**top**) and the simple DIC method (**bottom**).

*2.7. Photocatalytic Degradation Experiment*

As shown in Figure 8, the properties of the composites in terms of both adsorption and degradation were evaluated by comparing the performance of bio-template carbon, $MoS_2$, and the composites synthesized by two methods in dark and photocatalytic reactions. A blank group of pure humic acid solutions under the same conditions was introduced throughout the experiment to avoid experimental errors due to the self-degradation of humic acid caused by room temperature and visible light irradiation. In the dark reaction (−90–0 min), each material mainly underwent adsorption processes, and all materials were able to reach the dynamic equilibrium of adsorption within 60 min. The single-component materials were limited by the specific surface, and the adsorption capacity of pristine bio-template carbon (23.17%) and $MoS_2$ (13.03%) on the humic acid solution was much less than that of the composites obtained by the DIC (31.02%) and IHC methods (31.98%). It was inferred that the materials prepared by the DIC and IHC methods could make full use of the properties of the two components to synergistically enhance the adsorption capacity of the composite system. The difference between the materials became more significant after entering the photocatalytic reaction stage (0–210 min). The bio-template carbon only had adsorption properties, so the single carbon material could not degrade the humic acid solution under visible light irradiation during the photocatalytic reaction. Unlike bio-template carbon, the single $MoS_2$ photocatalytic semiconductor material exhibited the excellent degradation ability of humic acid driven by visible light. The catalytic degradation of humic acid by $MoS_2$ material reached the maximum value (48.75%) after 90 min of light irradiation. However, the photocatalytic performance of the material was limited

by the presence of severe photogenerated carrier recombination in pure $MoS_2$, and the degradation of humic acid was almost blocked in the subsequent reaction. Compared with pure materials, the photocatalytic degradation performance of the composites was much better than that of single-component materials. The introduction of bio-template carbon into the system increased the adsorption performance of the composites and, by forming a stable bond with MoS2, some photogenerated electrons in the photocatalytic semiconductor can be effectively transferred to the carbon material to achieve the effective separation of photogenerated carriers, further enhancing the photocatalytic degradation efficiency of composites. Among the two composites, the adsorption-catalysis effect of the material synthesized by the IHC method was better than that of the DIC method. The contact area of the photocatalytic material with the reactants was limited because the $MoS_2$ obtained by the DIC method was one-dimensional and needle-like. In addition, due to the difference in synthesis methods, the MoS2 synthesized by the DIC method was less crystalline than that by the HIC method, so the maximum catalytic effect of the composite system was about 92.03%, while the $MoS_2$, synthesized by the IHC method was in a two-dimensional petal shape and the contact area between the material and the solution was larger, making the photocatalytic degradation effect more significant. The adsorption degradation of humic acid by the composites reached 98.73% after 150 min of light illumination.

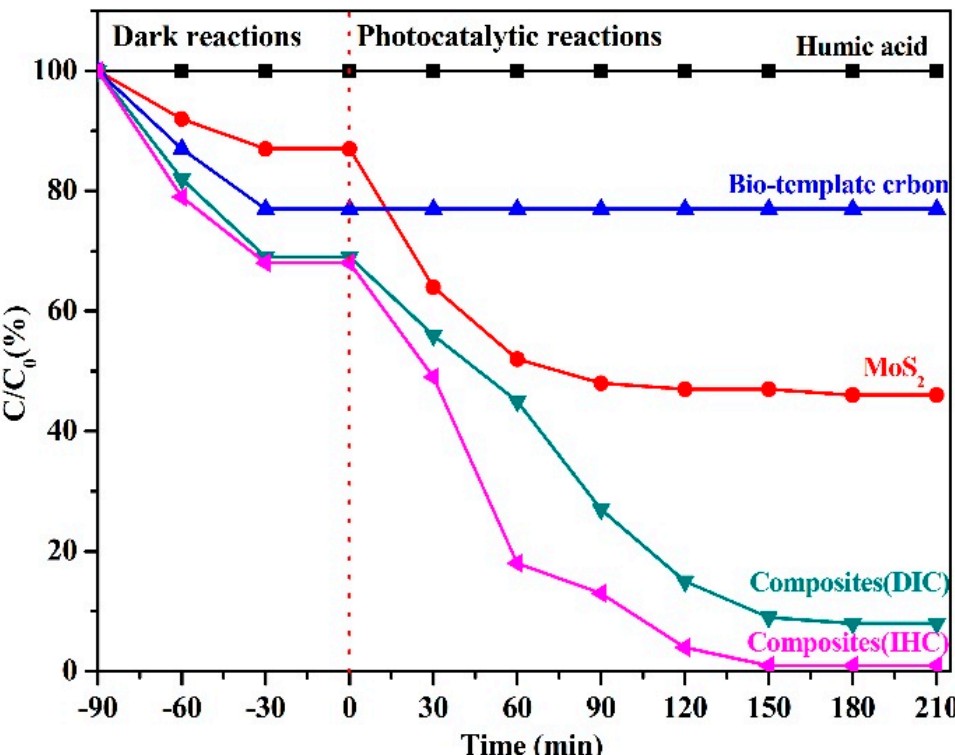

**Figure 8.** Degradation performance of bio-template carbon, pure $MoS_2$, and the composites synthesized by two methods for humic acid.

Based on the above results, we also conducted three cycles of the material to investigate the recyclability and photocatalytic stability of the material (Figure 9). After three cycles, the catalytic efficiency of the $MoS_2$/biocarbon composites prepared by the IHC method was about 96% of that in the first cycle. The catalytic efficiencies of the composites obtained by the pure biocarbon, $MoS_2$, and DIC methods all showed significant decreasing trends, with the catalytic efficiencies decreasing to 31%, 61%, and 63% of those at the first cycle, respectively. For the single biocarbon material, the main reason for the decrease in performance with an increasing number of cycles was due to the enrichment of humic acid molecules in the pore channels of the carbon material, which caused a blockage and thus

could not adsorb more molecules; for the single MoS$_2$ material, the decrease in material performance was mainly due to the rapid compounding of photogenerated carriers; and for the composites prepared by the DIC method, the decrease in catalytic efficiency was mainly due to the rapid compounding of MoS$_2$ and biocarbon contact mode restriction, although a part of photogenerated carriers was able to achieve effective separation, and with an increase in the number of cycles, the separation efficiency of carriers was significantly reduced, which had a serious impact on the catalytic efficiency. Combining the results of degradation and cycling experiments, the MoS$_2$/biocarbon composites prepared by the IHC method not only have outstanding photocatalytic degradation of humic acid but also have good recyclability and stability.

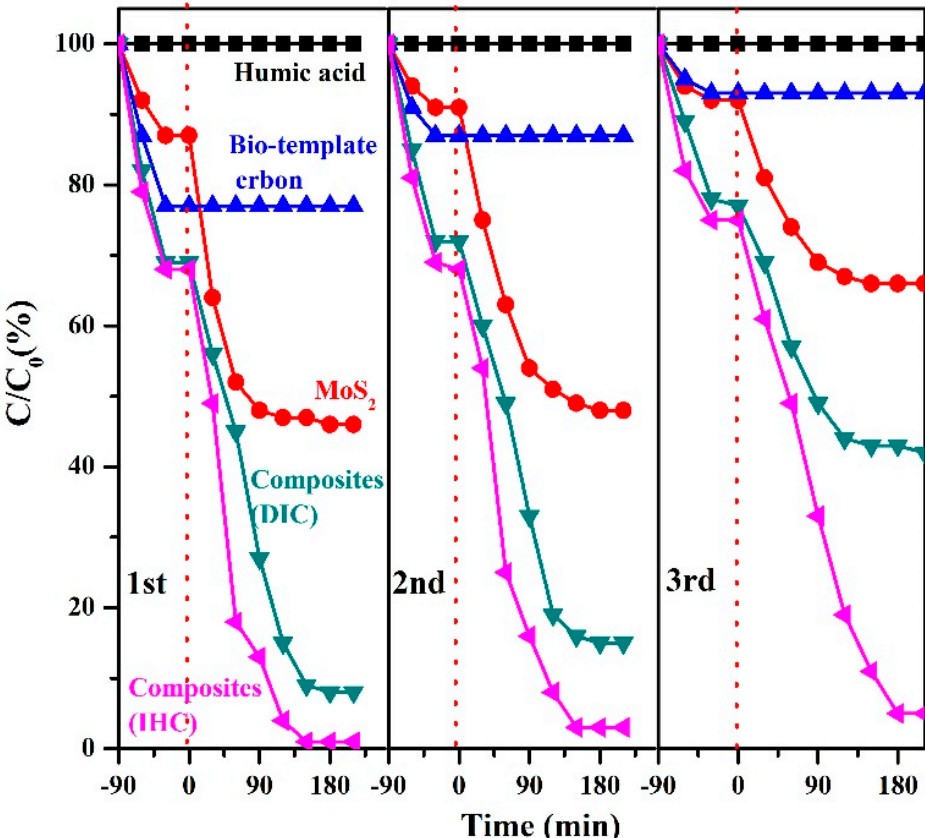

**Figure 9.** Catalytic efficiencies of the composites for three cycles obtained by the pure biocarbon, MoS$_2$, and DIC.

## 3. Materials and Methods

### 3.1. Chemical and Reagents

Ethanol (Analytical Reagent), sodium molybdate (Analytical Reagent), and thiourea (Analytical Reagent) were purchased from Sinopharm Chemical Reagent Co., Ltd. (Shanghai, China). and used without any further purification.

### 3.2. Synthesis Process

3.2.1. Pretreatment of Bio-Template

In order to remove experimentally useless impurities in leaves such as organelles, pigments, and dust particles, the bamboo leaves were soaked in ethanol for 72 h. The pretreated leaves were washed 3 times with de-ionized water and dried at room temperature.

3.2.2. Synthesis of Composites by Direct-Impregnation-Calcination (DIC) Method

The dried leaves were immersed in a mixed 0.1 mol·L$^{-1}$ solution of sodium molybdate, thiourea and de-ionized water was prepared, and the solution was left for another 72 h to obtain precursors. The precursor was calcined to 700 °C in nitrogen atmosphere and incubated for 2 h. The sheet-like solid MoS$_2$/biocarbon photocatalytic composite materials synthesized by direct-impregnation–calcination (DIC) method was prepared after cooling down to the room temperature.

3.2.3. Synthesis of Composites by Impregnation–Hydrothermal–Calcination (IHC) Method

The pretreated leaves from Step 2.2.1 and the configured 0.1 mol·L$^{-1}$ ionic solution were transferred to a Teflon inner container, which was equipped with a stainless steel outer casing and the whole reactor was heated at 220 °C for 24 h. After vacuum filtration and drying, lamellate leaves loaded with S and Mo ions obtained by hydrothermal method were anaerobic heated to the target temperature (decided by the analysis of TG-DSC test) in nitrogen atmosphere and incubated for 1 h. After cooled to the room temperature, the sheet-like solid MoS$_2$/biocarbon photocatalytic composite materials synthesized using the impregnation–hydrothermal–calcination (IHC) method were obtained [32].

*3.3. Characterization*

The micro-surface morphology analysis and phase structure of materials were examined using a Hitachi S-4800 field emission scanning electron microscopy (FESEM) and JEM 2010F transmission electron microscopy (JEOL, Japan), respectively. The crystal structures of materials were investigated by Bruker D8 Advance X-ray diffractometer with Cu K$\alpha$ radiation ($\lambda$ = 1.54056 Å), and the scanning step was 0.02 °·min$^{-1}$. The Raman spectra were obtained by DXR Microscope Raman spectroscopic microscope (Thermo Scientific, United States), with a wavelength of the test light source at 532 nm, and the scanning ranged from 10 to 3000 cm$^{-1}$. The valence state and bonding mode of each element in the material were obtained by measuring and analyzing the corresponding energy spectrum using a ESCALAB250 X-ray photoelectron spectrometer(Thermo Scientific- Scientific Instruments and Aut, United Kingdom).

*3.4. Photocatalytic Degradation Experiment*

The catalytic performance of the material was assessed by the visible light-driven degradation of humic acid in an XPA-II photochemical reactor (Xujiang machine electronic plant, Nanjing, China). The photocatalytic degradation experiments were carried out as follows: 100 mg of material was weighed and added to a reactor containing 100 mL of humic acid solution (concentration: 100 ppm). The reactor was installed in the instrument rack, and the mixture was magnetically stirred in the dark for about 90 min to fully operate the dark reaction experiment. After the dark reaction was completed, the magnetic stirring was continued, and the reactor was illuminated with a visible light source simulated by a Xenon lamp(Beijing Perfectlight Technology Co., Ltd., Beijing, China) to drive the photocatalytic reaction. During the photocatalytic reaction, cold water should be inserted into the reactor shell to eliminate the result interference caused by the degradation of humic acid due to thermal reaction (As shown in Figure S1). Throughout the experiment, 5 mL of solution was taken from the reactor at 30 min intervals and UV–visible absorption parameters of the solution were measured after high-speed centrifugation until the value did not change significantly. The photocatalytic degradation of humic acid was calculated as C/C$_0$, where C$_0$ was the absorbance of humic acid at the initial concentration (100 ppm) and C was the absorbance of humic acid at this moment.

**4. Conclusions**

In general, MoS$_2$/bio-template carbon photocatalytic composites were successfully synthesized through a novel impregnation–hydrothermal–calcination (IHC) method. Compared with same composites prepared by the direct-impregnation–calcination (DIC) method,

MoS$_2$ in the materials was in a petal-shaped and formed stable bonds with bio-template carbon. The bonds formed by the supported sulfides and carbon substrates are beneficial for the efficient separation of some photogenerated carriers in photocatalytic semiconductors. The experimental results of the photocatalytic degradation of humic acid further confirmed that the better performance of the adsorption and catalytic degradation from IHC method. After being illuminated for 150 min, the degradation of humic acid by the material reached 98.73%, achieving the low-cost and high-efficiency degradation of the humic acid aqueous solution using visible light under normal temperature and pressure. The experimental results provided a new possibility for semiconductor photocatalytic materials in the field of degradation applications.

**Supplementary Materials:** The following supporting information can be downloaded at: https://www.mdpi.com/article/10.3390/catal12111423/s1, Figure S1: Diagrammatic sketch of diagrammatic sketch.

**Author Contributions:** Conceptualization, H.N.; software, J.C.; formal analysis, J.W.; writing—original draft preparation, C.W.; writing—review and editing, N.W. and J.Q.; visualization, C.Y.; supervision, Z.W.; project administration, J.Q. All authors have read and agreed to the published version of the manuscript.

**Funding:** This research was funded by the Natural Science Foundation of Jiangsu Province (Grants No. BK20180971); Jiangsu Collaborative Innovation Center of Technology and Material for Water Treatment (XTCXSZ2019-1); Qinghai Province Key R&D and Transformation Program (2022-SF-137); Technology Development Project of Changshu (CS202008); Jiangsu Key Laboratory for Environment Functional Materials (SJHG1802).

**Data Availability Statement:** Data available on request from the authors.

**Conflicts of Interest:** The authors declare no conflict of interest.

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
