# Peer review of "Construction and Synthesis of MoS2/Biocarbon Composites for Efficient Visible Light-Driven Catalytic Degradation of Humic Acid"

_catalysts, doi:10.3390/catal12111423_

Round 1

Reviewer 1 Report

This manuscript is well written and presented the experimental evidence.  However, the authors must answer why the composites perform better than the semiconductor MoS2 for a photodegradation process. They should calculate the tauc plot to estimate the band gaps of MoS2 and the composites and compare them.  Secondly, authors should also perform photodegradation experiment to assess the stability and reusability of the composites. 

Author Response

Dear editor,

Thank you for your letter and for the reviewers’ comments concerning our manuscript entitled "catalysts-2003015". We sincerely appreciate the valuable feedback from you and referees. We have studied comments carefully and hope that the correction will meet with approval. The point-by-point answers to the comments and suggestions were listed as below. The reviewer comments are laid out below in italicized font and specific concerns have been numbered. Our response is given in normal font and changes/additions to the manuscript are given in red text.

# Reviewer 1

  1. However, the authors must answer why the composites perform better than the semiconductor MoS2 for a photodegradation process.

Response: Thank you for your question. The reason for the superior photodegradation of composites over semiconductor MoS2 is analyzed in 3.7, as follows “However, the photocatalytic performance of the material was limited by the presence of severe photogenerated carrier recombination in pure MoS2, and the degradation of humic acid was almost blocked in the subsequent reaction. Compared with pure materials, the photocatalytic degradation performance of the composites was much better than that of single-component materials. The introduction of bio-template carbon into the system increased the adsorption performance of the composites, and by forming a stable bond with MoS2, some photogenerated electrons in the photocatalytic semiconductor can be effectively transferred to the carbon material to achieve the effective separation of photogenerated carriers, further enhancing the photocatalytic degradation efficiency of composites”.

  1. They should calculate the tauc plot to estimate the band gaps of MoS2 and the composites and compare them.

Response: In fact, the effect of carbon is to add a defect energy level that leads to a raised adsorption background in the full band in the UV diagram. Therefore, the change of the forbidden band is not obvious in the actual tauc plot. This phenomenon is present in much of the photocatalytic literature. When we calculated the band gaps of MoS2 and the composites by tauc plot for comparison, the band gaps of the MoS2 semiconductor material and the two composites were almost the same as follows: please see attachment

Therefore, we have not included this image in the text. We believe that the introduction of bio-template carbon into the system increased the adsorption performance of the composites, and by forming a stable bond with MoS2, some photogenerated electrons in the photocatalytic semiconductor can be effectively transferred to the carbon material to achieve the effective separation of photogenerated carriers, further enhancing the photocatalytic degradation efficiency of composites.

  1. Secondly, authors should also perform photodegradation experiment to assess the stability and reusability of the composites.

Response: Thank you for your suggestion. In fact, we conducted the original photocatalytic degradation effect study along with the cyclability and stability of the material. We have included a discussion of these results at the end of 3.7.

Reviewer 2 Report

Comments:

This work (catalysts-1989310; Construction and Synthesis of MoS2/Biocarbon Composites for Efficient Visible Light-driven Catalytic Degradation of Humic Acid) reported the MoS2/bio-template carbon composite materials with outstanding photocatalytic degradation performance, synthesized by an impregnation-hydrothermal-calcination method. Composites of the same type were synthesized by a direct-impregnation-calcination method for comparison. Also claim that the calcination process could obtain from bio template carbon with preserved structure. I recommend the authors should address the following "minor issues" before the paper is accepted for publication in "catalysts" journal:

·       In “2.2. Synthesis process,direct-impregnation-calcination authors used 700C in the presence of nitrogen, what calcination at this high temperature? What form was the leaves, solid, powder etc? Also, author mentioned that “The resulting material was prepared after cooling down to room temperature”. What does it mean by “resulting material was prepared”.?  Need to clarify and should rewrite these line.

·       Also need to mention the form of material obtained after experiments, such as solid, powder, etc in the “2.2.3. Synthesis of composites by impregnation-hydrothermal-calcination (IHC) method” part. Can take help from the latest article; Materials Chemistry and Physics 278 (2022): 125640

·       On page 3 (line135) “..cold  water should be inserted into the reactor shell to eliminate the result interference caused by the degradation of humic acid due to thermal reaction.” How do authors insert the cold water.? is the author adding the cold water into the reaction? Or is this cold water in the outer jacket of the reaction chamber to cool it down? Authors need to clarify; it is a confusing statement. Must need to rewrite and can add the reactor picture into supplementary data with labels.

·       How authors measured the weight losses such as on page 4 (line154. Line 158, line 162) “The first stage of weight loss was about 15% / 42% weight loss consisted of two simultaneous / reason for the 6% weight loss of the system after 500 °C.? these weight losses are accurate measured?

·       On page 5 (line196), “The SEM images of materials obtained by the two methods” should rewrite it so it can make sense.it is confusing; does the author use two methods for SEM images or materials obtained by two methods? need to rewrite and mention the material name instead of material/methods.

·       In figure 5 in the XRD pattern, DIC shows some extra peaks (between 55° to 60°, between 20° to 30°), and having high intensity. Why? Label them correctly can take help from these references; (Chemical Physics Letters, 753, 137604) and (Advanced Powder Technology, 29(12), 3233-3240).

·       The degradation performance in figure-8 indicated that at 150, 180, and 210 min, the IHC showed performance, is the degradation performance value at this time frame is zero, meaning IHC degraded the humic acid completely?

Author Response

Dear editor,

Thank you for your letter and for the reviewers’ comments concerning our manuscript entitled "catalysts-2003015". We sincerely appreciate the valuable feedback from you and referees. We have studied comments carefully and hope that the correction will meet with approval. The point-by-point answers to the comments and suggestions were listed as below. The reviewer comments are laid out below in italicized font and specific concerns have been numbered. Our response is given in normal font and changes/additions to the manuscript are given in red text.

#Reviewer 2

  1. In “2.2. Synthesis process,” direct-impregnation-calcination authors used 700C in the presence of nitrogen, what calcination at this high temperature? What form was the leaves, solid, powder etc? Also, author mentioned that “The resulting material was prepared after cooling down to room temperature”. What does it mean by “resulting material was prepared”.? Need to clarify and should rewrite these line.

Response: Thank you for your question. For the direct-impregnation-calcination method, the leaves are transferred to a tube furnace after the impregnation process and are heated adiabatically in a N2 atmosphere. During the transformation of the leaf from template to biocarbon, the leaf remains intact as a sheet (solid). The term "resulting material" refers to MoS2/biocarbon photocatalytic composite materials prepared by the DIC method. As you mentioned, our previous expression was ambiguous, so we have rewritten parts of 2.2 where was hard for readers to understand.

  1. Also need to mention the form of material obtained after experiments, such as solid, powder, etc in the “2.2.3. Synthesis of composites by impregnation-hydrothermal-calcination (IHC) method” part. Can take help from the latest article; Materials Chemistry and Physics 278 (2022): 125640.

Response: Thank you for your suggestion. After reading Materials Chemistry and Physics 278 (2022): 125640, we found that the description of the material form is effective in helping the reader to understand more intuitively what we are studying. Therefore, we have rewritten 2.2.2 and 2.2.3 and referenced the article "Materials Chemistry and Physics 278 (2022): 125640

  1. On page 3 (line135) “..cold water should be inserted into the reactor shell to eliminate the result interference caused by the degradation of humic acid due to thermal reaction.” How do authors insert the cold water.? is the author adding the cold water into the reaction? Or is this cold water in the outer jacket of the reaction chamber to cool it down? Authors need to clarify; it is a confusing statement. Must need to rewrite and can add the reactor picture into supplementary data with labels.

Response: Thank you for your question. As you said, the cold water is not injected directly into the reaction solution. The reactor is a double layer transparent shell construction. The reactor has an interface at the top and bottom of the reactor, one end of which is fixed to the outer shell of the reactor and the other end is connected to a skin tube to pass cold water in and out. The cold water enters the sandwich space between the inner and outer shells through the lower interface and exits through the upper interface after absorbing heat. To avoid misleading the reader's understanding, we have rewritten the expressions in the text and added schematics and labels to the supporting information.

  1. How authors measured the weight losses such as on page 4 (line154. Line 158, line 162) “The first stage of weight loss was about 15% / 42% weight loss consisted of two simultaneous / reason for the 6% weight loss of the system after 500 °C.? these weight losses are accurate measured?

Response: Thank you for your question. In fact, we analyze the TG (Mass) data by segmenting the DCS according to where the peaks and inflection points occur, and these are automatically calculated and read by the software's mathematical program. Then the horizontal coordinates (temperature) of the segmented end points are back-propagated to the TG curve to find the corresponding mass, and finally the mass is subtracted to get the exact value.

  1. On page 5 (line196), “The SEM images of materials obtained by the two methods” should rewrite it so it can make sense.it is confusing; does the author use two methods for SEM images or materials obtained by two methods? need to rewrite and mention the material name instead of material/methods.

Response: Thank you for your suggestion. Indeed, the original expression was not easy for the reader to understand, so we have rewritten and replaced the content of line 196. The new content is "Fig.3 displayed SEM figures of MoS2/biocarbon photocatalytic composite materials obtained by the direct-impregnation-calcination (DIC) method and impregnation-hydrothermal-calcination (IHC) method respectively."

  1. In figure 5 in the XRD pattern, DIC shows some extra peaks (between 55° to 60°, between 20° to 30°), and having high intensity. Why? Label them correctly can take help from these references; (Chemical Physics Letters, 753, 137604) and (Advanced Powder Technology, 29(12), 3233-3240).

Response: Thank you for the suggestion. Through reading Chemical Physics Letters, 753, 137604 and Advanced Powder Technology, 29(12), 3233-3240, we found that the additional peaks appearing at 20-30° in the XRD of the materials prepared by the DIC method are mainly caused by the amorphous biocarbon. And the peaks appearing at 55-60° are the (110) and offset (008) crystallographic planes of MoS2, which we have marked in the figure in order to ensure that the reader can interpret the XRD accurately. In addition, because Chemical Physics Letters, 753, 137604 and Advanced Powder Technology, 29(12), 3233-3240 is very helpful and informative for our analysis of XRD, we have referenced and cited it in our paper.

  1. The degradation performance in figure-8 indicated that at 150, 180, and 210 min, the IHC showed performance, is the degradation performance value at this time frame is zero, meaning IHC degraded the humic acid completely?

Response: Thank you for your question. Your understanding is correct, when the photodegradation proceeds to 150 min, the photocatalytic degradation effect curve of the material obtained by the IHC method remains unchanged. At this point, the corresponding value of C/C0 is less than 1%, so it can be considered that the material obtained by the IHC method almost completely degrades humic acid.

We tried our best to improve the manuscript. We appreciate for Editors/Reviewers’ warm work earnestly, and hope that the correction will meet with approval.

Once again, thank you very much for your comments and suggestions.

Best wishes,

Junchao Qian